# Finite-time attitude tracking control for spacecraft based on backstepping method with input saturation

**Yanmin Ren**[1]*, **Aijia Xing**[2]

**1** Harbin University, Harbin, China, **2** Harbin University of Science and Technology, Harbin, China

\* renyanmin@hrbu.edu.cn

## Abstract

In this paper, the finite-time attitude tracking control problem for spacecraft based on the backstepping method is addressed. Firstly, a finite-time controller is designed, which can provide robustness for external disturbance by employing an improved adaptation law. Secondly, a novel finite-time controller with input saturation is proposed by introducing the hyperbolic function and auxiliary system to guarantee the control torques below a predetermined value. The above two controllers are continuous, thus, they are chattering-free. Finally, simulation results are presented to illustrate the effectiveness of the control strategies.

## 1 Introduction

In recent years, the attitude control problem of spacecraft has gained extensive attention. This has been motivated by the benefits gained via its space applications, such as space rendezvous and docking, spacecraft formation flying, deep space exploration, etc. Recently, various nonlinear control methods have been proposed for solving the attitude tracking control problem of spacecraft, such as adaptive sliding film attitude control [1], asymptotic tracking control [2], hybrid attitude saturation and fault-tolerant control [3], Quaternion-Based Hybrid Control [4], robust control [5,6], arbitrary perturbation control based on Tan-type BLFs [7], adaptive control based on filtered concurrent learning [8], and decentralized coordinated [9,10] control for spacecraft formation flying, etc. In reference [11], for the input amplitude/rate constraints of unmanned aerial systems, the algorithm is constructed using a saturation function and a damping term, relying on Barbalat's Lemma and graph theory to achieve group synergy, but the control objective is only asymptotically stable, and does not achieve the finite-time convergence characteristic. The above attitude control laws are asymptotically stable, which means tracking errors of closed-loop systems converge to equilibrium as time goes to infinity. Compared with asymptotic control methods, the finite-time control method can provide faster convergence and higher control precision. Thus, it is applied in spacecraft attitude tracking control widely.

Up to now, the existing finite-time control strategies mainly contain two categories: the homogeneous theory approach and the Lyapunov-based approach. In reference [12], the homogeneous theory is used to propose a finite-time control algorithm for spacecraft

**Data availability statement:** "All relevant data are within the paper and its Supporting Information files."

**Funding:** The author(s) received no specific funding for this work.

**Competing interests:** The authors have declared that no competing interests exist.

formation flying. References [13] and [14] propose robust and fault-tolerant finite-time control methods for spacecraft attitude control by using sliding mode control. In reference [15], finite-time control methods based on fast terminal sliding mode control are derived for spacecraft attitude synchronization and tracking. Reference [16] developed a finite-time output feedback attitude control method by adding a power integrator technique. The rigid spacecraft system is a standard cascade system and the backstepping method is a great cascade design tool, so the backstepping method [17,18] has been applied in finite-time attitude tracking control of rigid spacecraft successfully. Although the above methods solve the saturation problem, they face limitations. For example, reference [19] focuses on robust and consistent control of second-order uncertain multi-intelligent body systems, which is based on the fixed-time stability theory and Lyapunov function to deal with the speed and input constraints, but its controllers introduce discontinuities due to the sgn() function and the segmented saturation function, which may lead to jittering problems in the real system.

It is worthwhile to mention that to guarantee a fast convergence rate, the finite-time control output always reaches a large magnitude, especially within the initial phase of system response. Thus, many researchers have made effort in the study of finite-time control methods with actuator saturation. The homogeneous theory and saturated function in reference [20] are used to design a bounded proportional-derivative-type finite-time attitude control law. In order to deal with external disturbances, the finite-time observer [21] is adopted to develop a finite-time controller via output feedback with the adaptive method to ensure bounded input. The fast terminal sliding mode and Chebyshev neural network [22] are employed to derive a bounded finite-time distributed cooperative attitude control law for multiple spacecrafts. So far, although many finite-time controllers with actuator saturation have been proposed, there are few results on bounded finite-time controllers based on the backstepping method. References [23–26] use the backstepping technique to design controllers for spacecraft with actuator saturation, but the first three proposed control laws can not achieve finite-time stability and the last one is not continuous. References [11] and [19] further illustrate trade-offs between asymptotic stability and robustness under input/velocity constraints, emphasizing the need for continuous, chattering-free finite-time controllers.

Inspired by the realities stated above, this paper designs two finite-time attitude tracking controllers for the spacecraft by using the backstepping method. The main contributions of the results in this paper are: (i) The first controller can inhibit external disturbances requiring no information about them without chattering.(ii) The second controller can overcome the input saturation problem effectively in the presence of external disturbances with known bounds.

This paper is organized as follows. Sect 2 states the attitude tracking dynamics model of the spacecraft. In Sect 3, the design procedure of two continuous finite-time controllers is given in detail, and the stability of the closed-loop system is proved. A simulation example is then shown in Sect 4. Sect 5 concludes this paper.

## 2 Posture control model and problem description

### 2.1 Spacecraft attitude control model

The attitude, kinematics, and dynamics of a rigid spacecraft are modeled as

$$\dot{\mathbf{R}}_c = \mathbf{R}_c(\boldsymbol{\omega}_c)^{\times} \tag{1}$$

$$\mathbf{J}\dot{\boldsymbol{\omega}}_c + (\boldsymbol{\omega}_c)^{\times}\mathbf{J}\boldsymbol{\omega}_c = u + \mathbf{d} \tag{2}$$

$$(\boldsymbol{\omega}_c)^\times = \begin{bmatrix} 0 & -\omega_{c3} & \omega_{c2} \\ \omega_{c3} & 0 & -\omega_{c1} \\ -\omega_{c2} & \omega_{c1} & 0 \end{bmatrix} \tag{3}$$

where $\mathbf{R}_c \in \mathrm{SO}(3)$ is the rotation matrix that transforms the body frame to the inertial frame. $\boldsymbol{\omega}_c \in R^{3\times1}$ is the spacecraft's angular velocity. $u \in R^{3\times1}$ and $\mathbf{d} \in R^{3\times1}$ are the control torque and external disturbance torque, respectively. They are expressed in the body frame. $J \in R^{3\times3}$ is the spacecraft's inertial matrix. $(\boldsymbol{\omega}_c)^\times$ is the skew-symmetric matrix that transforms a vector in $R^{3\times1}$ to a $3 \times 3$ skew-symmetric matrix.

To address the attitude tracking problem, a time-varying target attitude trajectory is described by $\mathbf{R}_t \in \mathrm{SO}(3)$, the attitude kinematics is written as

$$\dot{\mathbf{R}}_t = \mathbf{R}_t(\boldsymbol{\omega}_t)^\times \tag{4}$$

where $\boldsymbol{\omega}_t \in R^{3\times1}$ is the target angular velocity.

The error rotation matrix is defined as

$$\mathbf{R}_e = \mathbf{R}_t^\mathrm{T}\mathbf{R}_c \tag{5}$$

The angular velocity error resolved in the body frame is introduced as

$$\boldsymbol{\omega}_e = \boldsymbol{\omega}_c - \mathbf{R}_e^T\boldsymbol{\omega}_t \tag{6}$$

As the error rotation matrix $\mathbf{R}_e$ can't be used to design the controller, basically, we use a new attitude error demonstrated [27]

$$\mathbf{e} = \frac{1}{2\sqrt{1 + \mathrm{tr}(\mathbf{R}_e)}}(\mathbf{R}_e - \mathbf{R}_e^T)^\vee \tag{7}$$

where the map $\vee$ denotes the inverse of the cross-product operation.

Then, based on the equation $\dot{\mathbf{R}}_e = \mathbf{R}_e(\boldsymbol{\omega}_e)^\times$, the attitude error kinematics and dynamics of the spacecraft are derived as follows.

$$\dot{e} = E\boldsymbol{\omega}_e \tag{8}$$

$$\mathbf{J}\dot{\boldsymbol{\omega}}_e = F + u + \mathbf{d} \tag{9}$$

$$F = -(\boldsymbol{\omega}_c)^\times\mathbf{J}\boldsymbol{\omega}_c - \mathbf{J}(\boldsymbol{\omega}_c)^\times\boldsymbol{\omega}_e - \mathbf{J}R_e^T\dot{\boldsymbol{\omega}}_t \tag{10}$$

$$\mathbf{E} = \frac{1}{2\sqrt{1 + \mathrm{tr}(\mathbf{R}_e)}}(\mathrm{tr}(\mathbf{R}_e^T)\mathbf{I}_{3\times3} - \mathbf{R}_e^T + 2\mathbf{e}\mathbf{e}^T) \tag{11}$$

Note that (Eqs 8) and (11) are singular, when the case $tr(\mathbf{R}_e) = -1$ occurs. So in order to solve this problem, the attitude error $\mathbf{e}$ and $E$ should be defined in the sublevel $L = \{R_e \in \mathrm{SO}(3) \mid \|\mathbf{e}\| < 1\}$.

## 2.2 Problem description

In this paper, a tracking spacecraft tracks a target spacecraft with a time-varying motion state, wherein the tracking spacecraft and the target spacecraft measure their own attitude and angular velocity through their respective measuring elements, and the attitude and angular velocity of the target spacecraft are transmitted to the tracking spacecraft in real time through

wireless communication. The tracking spacecraft uses its own state information and the state information of the target spacecraft to calculate the relative information of the rendezvous and docking.

The finite-time attitude tracking control problem of spacecraft studied in this paper can be described as: under the condition that the two spacecrafts of attitude tracking each measure their own state and the tracking spacecraft can obtain the state of the target spacecraft in real time, according to the attitude control model, the attitude tracking controller and the input saturation controller are designed using the finite-time control algorithm, the hyperbolic tangent function, and the auxiliary system. The relevant lemmas used in designing the controller are given below.

**Lemma 1** [28]. Suppose $\alpha_1, \alpha_2, \ldots, \alpha_n$ are all positive numbers and $0 < \rho < 2$, and then the following inequality holds

$$(\alpha_1^2 + \cdots + \alpha_n^2)^\rho \leq (\alpha_1^\rho + \cdots + \alpha_n^\rho)^2 \tag{12}$$

**Lemma 2** [29]. $V(x)$ is a continuous positive definite function. An extended Lyapunov description of finite-time stability can be given as

$$\dot{V}(\mathbf{x}) + \alpha V(\mathbf{x}) + \beta V(\mathbf{x})^\gamma \leq 0 \tag{13}$$

where $\alpha > 0$, $\beta > 0$, $0 < \gamma < 1$, and the converging time can be given as

$$T \leq \frac{1}{\alpha(1 - \gamma)} \ln \frac{\alpha V^{1-\gamma}(\mathbf{x}_0) + \beta}{\beta} \tag{14}$$

**Lemma 3** [15]. The inertia matrix $J$ is symmetric and positive, which is also bounded as $\lambda_{\min} x^T x \leq x^T J x \leq \lambda_{\max} x^T x$, where $x \in R^{3 \times 1}$, $\lambda_{\max}$ and $\lambda_{\min}$ are the maximum eigenvalue and minimum eigenvalue of $J$ respectively.

**Definition 1** [30]. For the following system $\dot{x} = f(x, u)$, where $x$ is the system state, $u$ is the control input. $x(t_0) = x(0)$, if there exist $\varepsilon$ and $T(\varepsilon, x(0)) < +\infty$, makes the inequality $\|x(t)\| \leq \varepsilon$ hold for $t \geq T + t_0$, then the system is practically finite-time stable.

**Lemma 4** [31]. For the above system, if there exists a continuous positive definite function $V(x)$, for real numbers $\alpha > 0$, $p \in (0, 1)$, $0 < \sigma < \infty$ and an open neighbourhood base $U \subset U_0$ containing the origin, a Lyapunov condition of finite-time stability can be given as $\dot{V} \leq -\alpha V^p + \sigma$, then the system is practically finite-time stable.

**Lemma 5** [32]. $V(x)$ is a continuous positive definite function, for any real number $c > 0$ and an open neighbourhood base $U \subset U_0$ containing the origin, a Lyapunov condition of finite-time stability can be given as $\dot{V} \leq -c$, and if $U = U_0 = R^n$, then the system is globally finite-time stable.

## 3 Design of the finite-time controller

### 3.1 Attitude tracking control with external disturbances

In this section, we assume that the external disturbances $d_i$, $i = 1, 2, 3$ are bounded, $d_{Mi} > 0$, $i = 1, 2, 3$ are all unknown constants, $\omega_t$ and $\dot{\omega}_t$ are bounded. Consider the rigid spacecraft system (8) and (9), the variables $x_1$ and $x_2$ are introduced as follows.

$$\mathbf{x}_1 = \mathbf{e}, \mathbf{x}_2 = \boldsymbol{\omega}_e - \boldsymbol{\omega}_e^v \tag{15}$$

In the light of (Eq 8), a virtual controller that makes the attitude error **e** always satisfy $\|\mathbf{e}\| < 1$ is proposed as

$$\boldsymbol{\omega}_e^v = -\beta_1 \mathbf{E}^{-1}\mathbf{x}_1 - \beta_2 \mathbf{E}^{-1}f(\mathbf{x}_1) + \mathbf{E}^{-1}\lambda \ln(1 - \mathbf{x}_1^T\mathbf{x}_1)\mathbf{x}_1 \tag{16}$$

$$f(x_{1,i}) = \begin{cases} r_1 x_{1,i} + r_2 \text{sign}(x_{1,i})x_{1,i}^2 & |x_{1,i}| \leq \eta, i = 1, 2, 3 \\ \text{sig}(x_{1,i})^\gamma & \text{otherwise} \end{cases} \tag{17}$$

where $f(\mathbf{x}_1) = [f(x_{1,1}), f(x_{1,2}), f(x_{1,3})]^T$, $\text{sig}(x_{1,i})^\gamma = \text{sign}(x_{1,i})|x_{1,i}|^\gamma, i = 1, 2, 3, \beta_1 > 0, \beta_2 > 0,$ $r_1 = (2 - \gamma)\eta^{\gamma-1}, r_2 = (\gamma - 1)\eta^{\gamma-2}, 0 < \gamma < 1, \eta > 0.$

**Proposition 1.** Consider (Eq 5), when $\boldsymbol{\omega}_t$ and $\dot{\boldsymbol{\omega}}_t$ are bounded, $x_{1,i}(i = 1, 2, 3)$ will converge to $|x_{1,i}| \leq \eta$ in finite time with the virtual controller (16).

**Proof.** Consider the following Lyapunov function.

$$V_{1,i} = \frac{1}{2}x_{1,i}^2, i = 1, 2, 3 \tag{18}$$

Its time derivative is $\dot{V}_{1,i} = x_{1,i}\dot{x}_{1,i}$, substituting (Eq 16) into it yields

$$\begin{aligned} \dot{V}_{1,i} &= -\beta_1 x_{1,i}^2 - \beta_2 x_{1,i}f(x_{1,i}) + \lambda \ln(1 - x_1^T x_1)x_{1,i}^2 \\ &\leq -\beta_1 x_{1,i}^2 - \beta_2 x_{1,i}f(x_{1,i}) \end{aligned} \tag{19}$$

Based on (Eq 17), using Lemma 1 obtains
When $|x_{1,i}| \leq \eta, i = 1, 2, 3$

$$\begin{aligned} \dot{V}_{1,i} &\leq -\beta_1 x_{1,i}^2 - \beta_2 r_1 x_{1,i}^2 - \beta_2 r_2 sign(x_{1,i})x_{1,i}^3 \\ &\leq -2(\beta_1 + \beta_2 r_1)V_{1,i} \end{aligned} \tag{20}$$

When $|x_{1,i}| > \eta, i = 1, 2, 3$

$$\begin{aligned} \dot{V}_{1,i} &\leq -\beta_1 x_{1,i}^2 - \beta_2 x_{1,i}sig(x_{1,i})^\gamma \\ &\leq -2\beta_1 V_{1,i} - 2^{(\gamma+1)/2}\beta_2 V_{1,i}^{(\gamma+1)/2} \end{aligned} \tag{21}$$

According to Lemma 2, it can be concluded that $x_{1,i}$ will converge to $|x_{1,i}| \leq \eta$ in finite time. Based on the virtual controller (16), a continuous controller is designed as

$$\mathbf{u} = -F + \mathbf{J}\dot{\boldsymbol{\omega}}_e^v - E^T x_1 - \mathrm{k}_1 x_2 - u_a x_2 \tag{22}$$

$$\dot{\hat{\psi}}_i = -\varepsilon_1 \hat{\psi}_i + \frac{1}{2}\mathrm{p}_1\chi^{-2}|\mathrm{x}_{2i}|^2 \tag{23}$$

where $u_a = \text{diag}(\mathrm{u}_{ai})$, $\mathrm{u}_{ai} = \frac{1}{2}\chi^{-2}\hat{\psi}_i$, $\hat{\psi}_i$ is the estimation of $\psi_i$, $\psi_i = \mathrm{d}_{\mathrm{Mi}}^2$, $\tilde{\psi}_i$ is the estimation error which is defined as $\tilde{\psi}_i = \psi_i - \hat{\psi}_i$, $\mathrm{k}_1 > 0, \varepsilon_1 > 0, \mathrm{p}_1 > 0, \chi > 0$.

The following Lyapunov function is chosen

$$V_2 = \frac{1}{2}\mathbf{x}_1^T x_1 + \frac{1}{2}\mathbf{x}_2^T J x_2 + \frac{1}{2}\sum_{i=1}^3 \frac{1}{p_1}\tilde{\psi}_i^2, \quad i = 1, 2, 3 \tag{24}$$

Its time derivative along the trajectory of the dynamical system (9) is

$$
\begin{aligned}
\dot{V}_2 = {} & -\beta_1 \mathbf{x}_1^T \mathbf{x}_1 - \beta_2 \mathbf{x}_1^T f(\mathbf{x}_1) + \lambda \ln(1 - \mathbf{x}_1^T \mathbf{x}_1) \mathbf{x}_1^T \mathbf{x}_1 + \\
& \mathbf{x}_1^T E \mathbf{x}_2 + \mathbf{x}_2^T (F + u + d - \mathbf{J} \dot{\omega}_e^\nu) - \sum_{i=1}^{3} \frac{1}{p_1} \tilde{\psi}_i \dot{\hat{\psi}}_i
\end{aligned}
\tag{25}
$$

Using Lemma 1 and Lemma 3, and substituting (Eq 22) and (Eq 23) into (Eq 25) obtain

$$
\begin{aligned}
\dot{V}_2 \leq {} & -\beta_1 \mathbf{x}_1^T \mathbf{x}_1 - k_1 \mathbf{x}_2^T x_2 - \frac{1}{2} \sum_{i=1}^{3} \frac{\varepsilon_1}{p_1} \tilde{\psi}_i^2 + \sum_{i=1}^{3} \frac{\chi^2}{2} + \frac{1}{2} \sum_{i=1}^{3} \frac{\varepsilon_1}{p_1} \psi_i^2 \\
\leq {} & -\beta_1 \frac{1}{2} \mathbf{x}_1^T \mathbf{x}_1 - \frac{k_1}{\lambda_{\max}} \frac{1}{2} \mathbf{x}_2^T J x_2 - \varepsilon_1 \frac{1}{2} \sum_{i=1}^{3} \frac{\tilde{\psi}_i^2}{p_1} + \sum_{i=1}^{3} \frac{\chi^2}{2} + \frac{1}{2} \sum_{i=1}^{3} \frac{\varepsilon_1}{p_1} \psi_i^2 \\
\leq {} & -\eta_1 V_2 + \varsigma_1
\end{aligned}
\tag{26}
$$

where $\eta_1 = \min(\beta_1, \frac{k_1}{\lambda_{\max}}, \varepsilon_1)$, $\varsigma_1 = \sum_{i=1}^{3} \frac{\chi^2}{2} + \frac{1}{2} \sum_{i=1}^{3} \frac{\varepsilon_1}{p_1} \psi_i^2$. As $\varsigma_1$ is bounded, based on the boundedness theorem, $\mathbf{x}_1$, $\mathbf{x}_2$ and $\tilde{\psi}_i$ are all uniformly ultimately bounded. In order to achieve the finite-time convergence, a finite-time controller is proposed as follows.

$$
\mathbf{u}_z = \mathbf{u} - k_2 \mathrm{sig}(\mathbf{x}_2)^\gamma
\tag{27}
$$

where $\mathrm{sig}(\mathbf{x}_2)^\gamma = \left[ |x_{2,i}|^\gamma \mathrm{sign}(x_{2,i}) \right]^T$, $i = 1, 2, 3$, $k_2 > 0$.

**Theorem 1**. Consider the spacecraft system described by (8) and (9), the following conclusions can be satisfied with the control law (27).

(i) $\mathbf{x}_1$ and $\mathbf{x}_2$ are practically finite-time stable.

(ii) $\omega_e$ is practically finite-time stable.

**Proof**. (i) Consider the Lyapunov function $V_2$, based on (Eq 26), its time derivative along the trajectory of the dynamical system (9) can be written as

When $|x_{1,i}| > \eta$, $i = 1, 2, 3$,

$$
\begin{aligned}
\dot{V}_2 \leq {} & -\beta_2 \mathbf{x}_1^T \mathrm{sig}(\mathbf{x}_1)^\gamma + \sum_{i=1}^{3} \frac{\psi_i}{2\chi^2} |x_{2i}|^2 + \sum_{i=1}^{3} \frac{\chi^2}{2} - k_2 \mathbf{x}_2^T \mathrm{sig}(\mathbf{x}_2)^\gamma \\
& - \sum_{i=1}^{3} \frac{1}{2\chi^2} \hat{\psi}_i |x_{2i}|^2 - \sum_{i=1}^{3} \frac{1}{p_1} \tilde{\psi}_i \dot{\hat{\psi}}_i \\
\leq {} & -\beta_2 \left( \frac{1}{2} \mathbf{x}_1^T \mathbf{x}_1 \right)^{\frac{\gamma+1}{2}} - \frac{k_2}{\lambda_{\max}^{(\gamma+1)/2}} \lambda_{\max}^{\frac{\gamma+1}{2}} \left( \frac{1}{2} \mathbf{x}_2^T x_2 \right)^{\frac{\gamma+1}{2}} - \varepsilon_1 \sum_{i=1}^{3} \left( \frac{1}{2} \frac{\tilde{\psi}_i^2}{p_1} \right)^{\frac{1-\gamma}{2}} \left( \frac{1}{2} \frac{\tilde{\psi}_i^2}{p_1} \right)^{\frac{\gamma+1}{2}} \\
& + \sum_{i=1}^{3} \frac{\chi^2}{2} + \frac{1}{2} \sum_{i=1}^{3} \frac{\varepsilon_1 \psi_i^2}{p_1} \\
\leq {} & -\beta_2 \left( \frac{1}{2} \mathbf{x}_1^T \mathbf{x}_1 \right)^{\frac{\gamma+1}{2}} - \frac{k_2}{\lambda_{\max}^{(\gamma+1)/2}} \left( \frac{1}{2} \mathbf{x}_2^T J x_2 \right)^{\frac{\gamma+1}{2}} - \varepsilon_1 a_{min} \sum_{i=1}^{3} \left( \frac{1}{2} \frac{\tilde{\psi}_i^2}{p_1} \right)^{\frac{\gamma+1}{2}} + \varsigma_1 \\
\leq {} & -\eta_2 V_2^{\frac{\gamma+1}{2}} + \varsigma_1
\end{aligned}
\tag{28}
$$

When $\eta_2 = \min(\beta_2, \frac{k_2}{\lambda_{\max}^{(\gamma+1)/2}}, \varepsilon_1 a_{min})$, $a_{min} = \min\left( \left( \frac{1}{2} \frac{\tilde{\psi}_i^2}{p_1} \right)^{(1-\gamma)/2}, \quad i = 1, 2, 3 \right)$.

When $|x_{1,i}| \leq \eta, i = 1, 2, 3$, $\mathbf{x}_1$ has it been in the convergence region $|x_{1,i}| \leq \eta, i = 1, 2, 3$

$$
\begin{aligned}
\dot{V}_2 &\leq -(\beta_1 + \beta_2 r_1)\mathbf{x}_1^T\mathbf{x}_1 + \sum_{i=1}^{3}\frac{\psi_i}{2\chi^2}|x_{2i}|^2 + \sum_{i=1}^{3}\frac{\chi^2}{2} - k_2\mathbf{x}_2^T\text{sig}(\mathbf{x}_2)^\gamma \\
&\quad - \sum_{i=1}^{3}\frac{\hat{\psi}_i|x_{2i}|^2}{2\chi^2} - \sum_{i=1}^{3}\frac{1}{p_1}\tilde{\psi}_i\dot{\hat{\psi}}_i \\
&\leq -(\beta_1 + \beta_2 r_1)\left(\tfrac{1}{2}\mathbf{x}_1^T\mathbf{x}_1\right)^{\frac{1-\gamma}{2}}\left(\tfrac{1}{2}\mathbf{x}_1^T\mathbf{x}_1\right)^{\frac{\gamma+1}{2}} + \varsigma_1 - \frac{k_2}{\lambda_{\max}^{(\gamma+1)/2}}\left(\tfrac{1}{2}\mathbf{x}_2^T J\mathbf{x}_2\right)^{\frac{\gamma+1}{2}} \\
&\quad -\varepsilon_1 a_{min}\sum_{i=1}^{3}\left(\tfrac{1}{2}\frac{\tilde{\psi}_i^2}{p_1}\right)^{\frac{\gamma+1}{2}} \\
&\leq -\eta_3 V_2^{\frac{\gamma+1}{2}} + \varsigma_1
\end{aligned}
\tag{29}
$$

where $\eta_3 = \min\left((\beta_1 + \beta_2 r_1)\left(\tfrac{1}{2}\mathbf{x}_1^T\mathbf{x}_1\right)^{\frac{1-\gamma}{2}}, \frac{k_2}{\lambda_{\max}^{(\gamma+1)/2}}, \varepsilon_1 a_{min}\right)$.

As $\varsigma_1$ is positive and bounded, according to the Lemma 4, $\mathbf{x}_1$ and $\mathbf{x}_2$ are all practically finite-time stable.

(ii) Based on (Eqs 15) and (16), the angular velocity error $\boldsymbol{\omega}_e$ can be written as follows

$$
\boldsymbol{\omega}_e = \mathbf{x}_2 - \beta_1\mathbf{E}^{-1}\mathbf{x}_1 - \beta_2\mathbf{E}^{-1}f(\mathbf{x}_1) + \mathbf{E}^{-1}\lambda\ln(1 - \mathbf{x}_1^T\mathbf{x}_1)\mathbf{x}_1
\tag{30}
$$

Furthermore, considering the finite-time convergence of $\mathbf{x}_1$ and $\mathbf{x}_2$, it can be concluded that $\boldsymbol{\omega}_e$ is also practically finite-time stable.

Now the proof has been completed.

**Remark 1**. The control law (27) is continuous and chattering-free. $\dot{\hat{\psi}}_i$ in (Eq 23) can not be strictly positive for all the time, which can avoid $\hat{\psi}_i$ growing without bound and reduce the control torque in a steady stage.

## 3.2 Attitude tracking control with external disturbances and input saturation

It should be noted that Sect 3.1 does not consider the input saturation problem, which can affect the control effect and stability of the system. In order to solve this problem, a novel finite-time controller with input saturation is proposed by introducing the hyperbolic function and auxiliary system in this section. We assume that the external disturbances $d$ are bounded, $\|d\| \leq d_{\max}$, $d_{\max}$ is a known positive constant, $\boldsymbol{\omega}_t$ and $\dot{\boldsymbol{\omega}}_t$ are bounded. The finite-time controller with input saturation is proposed as

$$
\mathbf{u} = -k_1\tanh(\varepsilon_1\varpi) - k_2\tanh(\varepsilon_2\mathbf{x}_1) - k_3\tanh(\varepsilon_3\dot{\mathbf{x}}_1)
\tag{31}
$$

$$
\delta = x_2 - \varpi
\tag{32}
$$

$$
\begin{aligned}
\dot{\varpi} &= \mathbf{J}^{-1}(F - \mathbf{J}\dot{\omega}_e^\nu + \mathbf{u}) + \left[\mathbf{x}_2^T(F - \mathbf{J}\dot{\omega}_e^\nu + \mathbf{u})\right. \\
&\quad \left. +\mathbf{x}_1^T\dot{\mathbf{x}}_1\right]\frac{\delta}{\delta^T\delta} + k_4\delta + \left(k_5 + d_{max}\|\mathbf{x}_2^T\|\right)\frac{\delta}{\delta^T\delta} + u_a
\end{aligned}
\tag{33}
$$

$$
u_a = \begin{cases} \frac{\delta}{\|\delta\|}\|\mathbf{J}^{-1}\|d_{\max} & \|\delta\| > n_1 \\ \frac{\text{sig}(\delta)^\gamma}{n_1^\gamma}\|\mathbf{J}^{-1}\|d_{\max} & \|\delta\| \leq n_1 \end{cases}
\tag{34}
$$

where $k_1 > 0$, $k_2 > 0$, $k_3 > 0$, $k_4 > 0$, $k_5 > 0$, $\varepsilon_1 > 0$, $\varepsilon_2 > 0$, $\varepsilon_3 > 0$, $n_1 > 0$.

**Theorem 2**. Consider the spacecraft system described by (8) and (9), the following conclusions can be satisfied with the control laws (31)–(34).

(i) When $\|\delta\| > n_1$, $\mathbf{x}_1$ and $\mathbf{x}_2$ will converge to equilibrium $\|\mathbf{x}_1\| = 0$ and $\|\mathbf{x}_2\| = 0$ in finite time, when $\|\delta\| \le n_1$, and $k_4$ satisfying $k_4 - \frac{(n_1^{\gamma} - \|\delta\|^{\gamma})\|\mathbf{J}^{-1}\|d_{max}}{n_1^{\gamma}\|\delta\|} \ge 0$, then $\mathbf{x}_1$ and $\mathbf{x}_2$ will converge to equilibrium $\|\mathbf{x}_1\| = 0$ and $\|\mathbf{x}_2\| = 0$ in finite time.

(ii) In finite time, $\boldsymbol{\omega}_e$ will converge to $\|\boldsymbol{\omega}_e\| = 0$.

**Proof.** (i) Consider the Lyapunov function $V_3 = \frac{1}{2}\mathbf{x}_1^T\mathbf{x}_1 + \frac{1}{2}\mathbf{x}_2^T\mathbf{J}\mathbf{x}_2 \quad + \quad \frac{1}{2}\delta^T\delta$, its time derivative along the trajectory of the dynamical system (9) can be written as

$$\dot{V}_3 = \mathbf{x}_1^T\dot{\mathbf{x}}_1 + \mathbf{x}_2^T(F - \mathbf{J}\dot{\boldsymbol{\omega}}_e^{\nu} + u + d) + \delta^T[\mathbf{J}^{-1}(F - \mathbf{J}\dot{\boldsymbol{\omega}}_e^{\nu} + u + d) - \dot{\varpi}]$$
$$= \delta^T\mathbf{J}^{-1}d + \mathbf{x}_2^Td - k_4\delta^T\delta - (k_5 + d_{max}\|\mathbf{x}_2^T\|) - \delta^Tu_a \tag{35}$$

When $\|\delta\| > n_1$,

$$\dot{V}_3 \le \|\mathbf{x}_2^T\|d_{max} + \|\delta^T\|\|\mathbf{J}^{-1}\|d_{max} - k_4\delta^T\delta - k_5 -$$
$$\|\mathbf{x}_2^T\|d_{max} - \|\delta\|\|\mathbf{J}^{-1}\|d_{max}$$
$$= -k_4\delta^T\delta - k_5 \tag{36}$$
$$\le -k_5$$

When $\|\delta\| \le n_1$, $k_4 \ge \frac{(n_1^{\gamma} - \|\delta\|^{\gamma})\|\mathbf{J}^{-1}\|d_{max}}{n_1^{\gamma}\|\delta\|}$,

$$\dot{V}_3 \le -k_4\delta^T\delta - k_5 + \|\delta^T\|\|\mathbf{J}^{-1}\|d_{max} - \frac{\|\delta\|^{\gamma+1}\|\mathbf{J}^{-1}\|d_{max}}{n_1^{\gamma}}$$
$$= -k_5 - k_4\delta^T\delta + \left(1 - \frac{\|\delta\|^{\gamma}}{n_1^{\gamma}}\right)\|\delta\|\|\mathbf{J}^{-1}\|d_{max}$$
$$= -k_5 - \left[k_4 - \frac{(n_1^{\gamma} - \|\delta\|^{\gamma})\|\mathbf{J}^{-1}\|d_{max}}{n_1^{\gamma}\|\delta\|}\right]\|\delta\|^2 \tag{37}$$
$$\le -k_5$$

Using Lemma 5 obtains that $V_3$ will converge to the region $V_3 = 0$ in finite time. Therefore, it can be concluded that $\mathbf{x}_1$ and $\mathbf{x}_2$ will converge to equilibrium $\|\mathbf{x}_1\| = 0$ and $\|\mathbf{x}_2\| = 0$ in finite time.

Based on (Eqs 15) and (16), considering the finite-time convergence of $\mathbf{x}_1$ and $\mathbf{x}_2$, it can be concluded that $\boldsymbol{\omega}_e$ will converge to the region $\|\boldsymbol{\omega}_e\| = 0$ in finite time.

Now the proof has been completed.

**Remark 2**. When $\|\delta\| = 0$, (Eq 33) will appear singular. To solve this problem, the term $\frac{\delta}{\delta^T\delta}$ is substituted by the following term

$$\text{sat}(\delta) = \begin{cases} \frac{\delta}{\delta^T\delta}, & \|\delta\| \ne 0 \\ \frac{\delta}{\delta^T\delta + \Delta}, & \|\delta\| = 0 \end{cases} \tag{38}$$

## 4 Simulation results

In this section, the proposed control schemes are applied to spacecraft attitude tracking control.

The inertia matrix and initial conditions of the spacecraft are:

$$\mathbf{J} = \begin{bmatrix} 22.7 & 0.3 & -0.2 \\ 0.3 & 23.5 & 0.5 \\ -0.2 & 0.5 & 24.6 \end{bmatrix} \text{kg} \cdot \text{m}^2, \mathbf{R}_c(0) = \begin{bmatrix} -0.5414 & -0.7072 & -0.4546 \\ -0.0009 & -0.5403 & 0.8415 \\ -0.8407 & 0.4560 & 0.2919 \end{bmatrix}, \tag{39}$$

$\omega_c(0) = \begin{bmatrix} 0.1 & 0.1 & 0.1 \end{bmatrix}^T$ rad/s.

The desired attitude and angular velocity for the spacecraft are: $\mathbf{R}_t(0) = I_3$, $\omega_t = -0.1\begin{bmatrix} \sin(t/40) & \sin(t/50) & \sin(t/60) \end{bmatrix}^T$ rad/s.

External disturbances are: $d = 0.002\begin{bmatrix} \sin(0.1t) & \cos(0.2t) & \sin(0.2t) \end{bmatrix}^T$ N.$m$.

Parameters of the controller (Eq 27) are selected as: $k_1 = 0.5$, $k_2 = 0.4$, $\beta_1 = 0.2$, $\beta_2 = 0.1$, $\eta = 0.0001$, $\lambda = 0.01$, $\gamma = 0.8$, $\varepsilon_1 = 0.05$, $p_1 = 0.01$, $\chi = 0.1$.

Figs 1–5 depict the performance of the controller (Eq 27). In the figures, $i$ represents the $i$ th element of the corresponding vector. It follows from Figs 1–3 that the attitude tracking maneuver can be achieved in 15 seconds. The control input is shown in Fig 4, where chattering is avoided because the controller can maintain continuity when dealing with external disturbances. The estimated parameters are depicted in Fig 5.

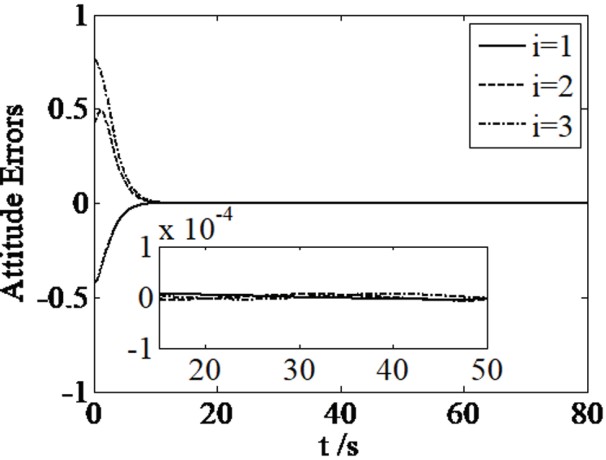

**Fig 1. Curves of attitude errors.**

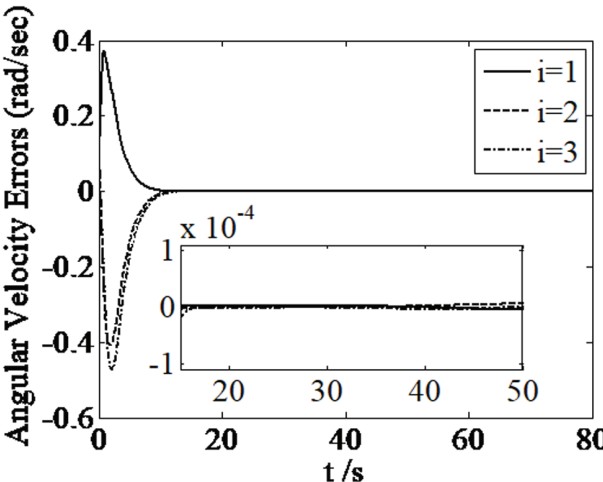

**Fig 2. Curves of angular velocity errors.**

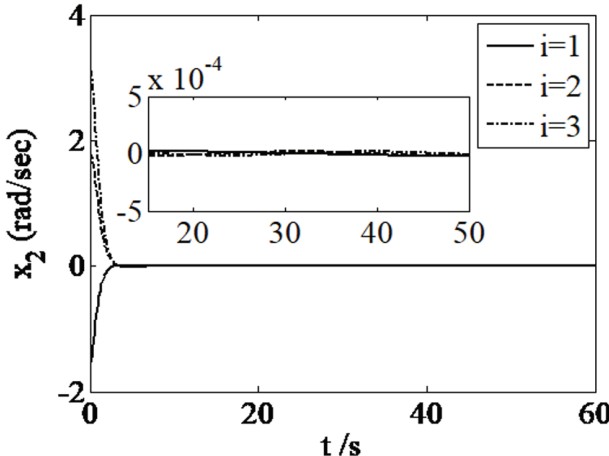

**Fig 3. Curves of $x_2$.**

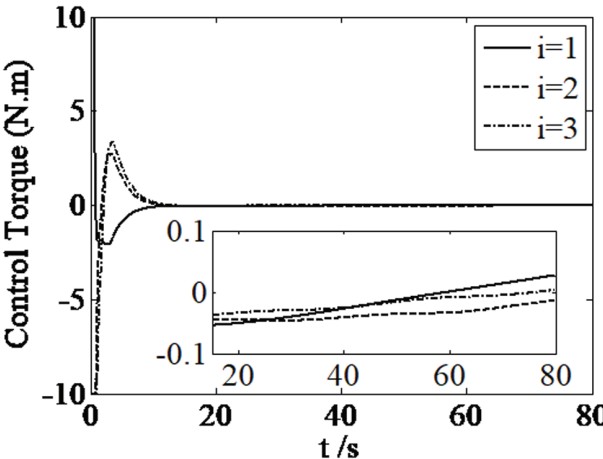

**Fig 4. Curves of control torque.**

In order to verify that the finite-time controller has faster convergence speed and higher steady-state accuracy, this paper will compare it with the finite-time controller based on the boundary layer theorem. In order to ensure the continuity of the controller, based on the boundary layer theorem, the controller is first designed as shown in (Eq 39).

$$\mathbf{u} = -F + \mathbf{J}_c \dot{\omega}_e^\nu - E^T x_1 - k_1 x_2 - u_a \tag{40}$$

where $u_a = [u_{a1}, u_{a2}, u_{a3}]^T$, $k_1 > 0$ and $n_1$ are small positive numbers, and $u_{a1}$, $u_{a2}$ and $u_{a3}$ are expressed by (Eq 40).

$$u_{ai} = \begin{cases} \frac{x_{2,i}}{|x_{2,i}|} \hat{d}_{Mi} & |x_{2,i}| \hat{d}_{Mi} > n_1 \\ \frac{x_{2,i}}{n_1} \hat{d}_{Mi}^2 & |x_{2,i}| \hat{d}_{Mi} \leq n_1 \end{cases} \tag{41}$$

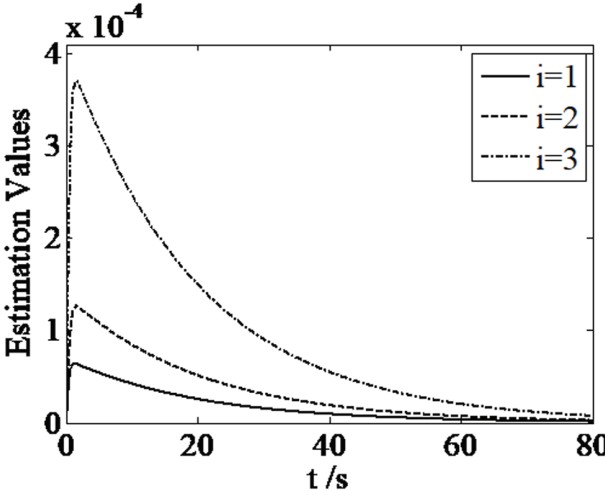

**Fig 5. Curves of estimated parameters.**

In order to solve the problem that the estimated value of the traditional adaptive update law may increase without limit, an improved adaptive update law is designed as shown in (Eq 41).

$$\dot{\hat{d}}_{Mi} = -\varepsilon\hat{d}_{Mi} + p\left|x_{2,i}\right|, \hat{d}_{Mi}(0) > 0 \tag{42}$$

where $\varepsilon > 0$, $p > 0$. According to the boundedness theorem, in order to ensure the finite time convergence of the system, the finite time controller is finally designed as (Eq 42).

$$\mathbf{u}_z = \mathbf{u} - k_2\mathrm{sig}(\mathbf{x}_2)^\gamma \tag{43}$$

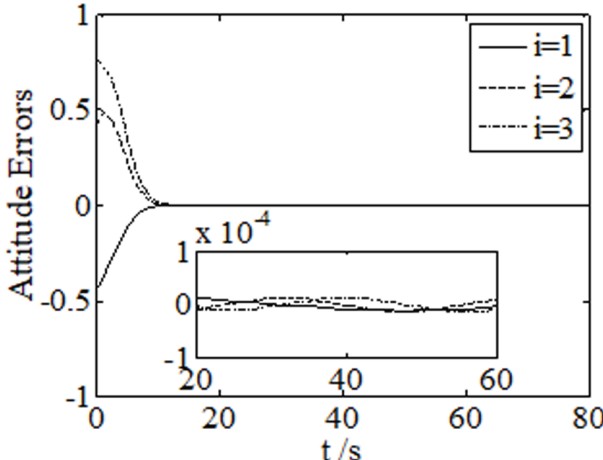

**Fig 6. Curves of attitude errors.**

where $\mathrm{sig}(\mathbf{x}_2)^\gamma = \left[\,|x_{2,i}|^\gamma \mathrm{sign}(x_{2,i})\,\right]^{\mathrm{T}}, i = 1, 2, 3, \mathrm{k}_2 > 0$. The specific control parameters are: $k_1 = 0.1$, $k_2 = 0.2$, $\beta_1 = 0.2$, $\beta_2 = 0.1$, $\gamma = 0.8$, $\eta = 0.0001$, $\mathrm{n}_1 = 0.0001$, $\lambda = 0.01$, $\varepsilon = 0.05$, $\mathrm{p} = 0.06$. The simulation results are shown in Figs 6–10.

It can be seen from Figs 6–8 that in the presence of external disturbances, the attitude error, angular velocity error, and achieve high-precision convergence within 20 seconds. In addition, it can be seen from the partial enlarged views of Figs 7 and 8 that the simulation curve in the steady-state stage is smooth and there is no jitter phenomenon. Comparing Fig 1 and Fig 2 with Fig 6 and Fig 7, it can be seen that the finite time controller designed in this paper has a faster convergence speed and higher steady-state accuracy. Fig 9 is a simulation curve of the control torque of the tracking spacecraft, and Fig 10 is a simulation curve of the upper limit estimate of the external disturbance torque.

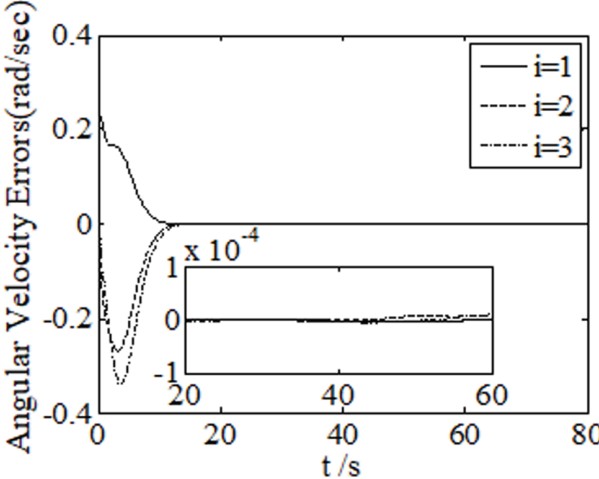

**Fig 7. Curves of angular velocity errors.**

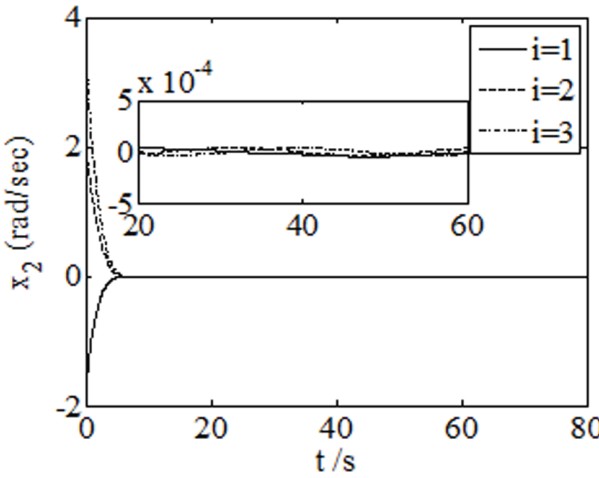

**Fig 8. Curves of $x_2$.**

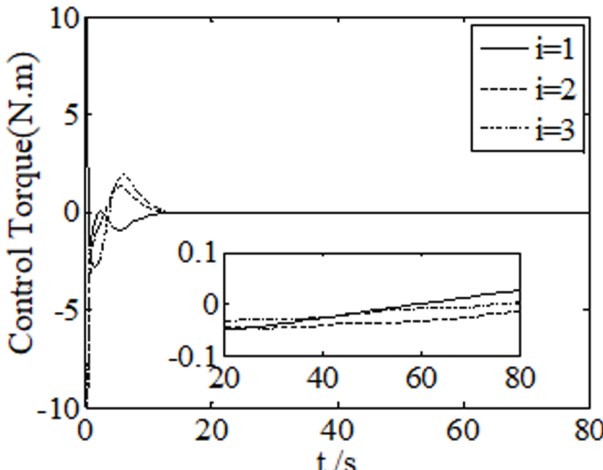

**Fig 9. Curves of control torque.**

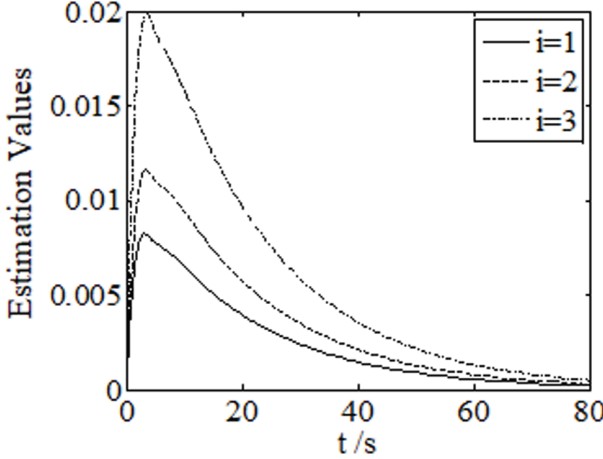

**Fig 10. Curves of estimated parameters.**

Parameters of the controller (Eq 31) are set as $k_1 = 1.5$, $k_2 = 1.5$, $k_3 = 1.5$, $k_4 = 7$, $k_5 = 0.002$, $\varepsilon_1 = 50$, $\varepsilon_2 = 50$, $\varepsilon_3 = 50$, $\beta_1 = 0.3$, $\beta_2 = 0.3$, $\eta = 0.001$, $n_1 = 0.001$, $\lambda = 0.001$, $\gamma = 0.8$. Figs 11–14 depict the performance of the controller (Eq 31). It follows from Figs 11–14 that the attitude tracking maneuver can be achieved in 20 seconds under the condition that the control input satisfying $u \leq 5N.m$.

Comparing Figs 1–4 and Figs 11–14, there is no remarkable change except that the controller (Eq 27) can provide a little faster convergence than the controller (Eq 31), so it concludes that the controller (Eq 31) can resolve the problem of actuator saturation well. The control input is shown in Fig 14, where chattering is avoided and the control input satisfies the physical limit of the actuators.

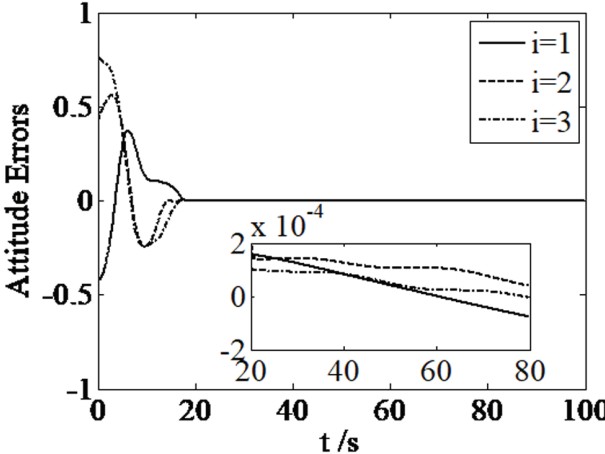

**Fig 11. Curves of attitude errors.**

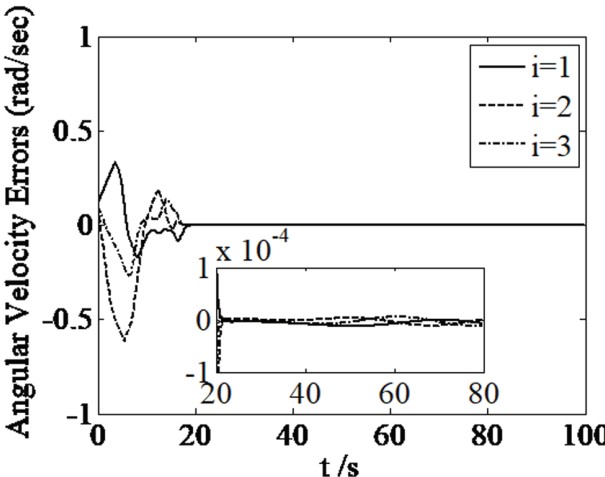

**Fig 12. Curves of angular velocity errors.**

## 5 Conclusion

In this paper, the finite-time control to achieve spacecraft attitude tracking based on the backstepping method is presented. The external disturbances, chattering, and input saturation are considered in the controller's design. The proposed controllers can prevent chattering phenomena and achieve finite-time tracking. In addition, the second controller has been demonstrated to have superior performance while still holding stability in the presence of input saturation. A simulation example is shown to support the above analysis and demonstrate excellent dynamic tracking performance using the proposed finite-time controllers. The results show that the presented controllers can achieve spacecraft attitude tracking accurately in finite time.

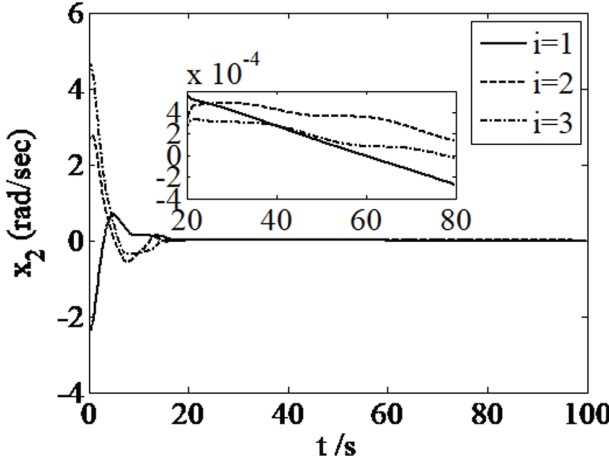

**Fig 13. Curves of $x_2$.**

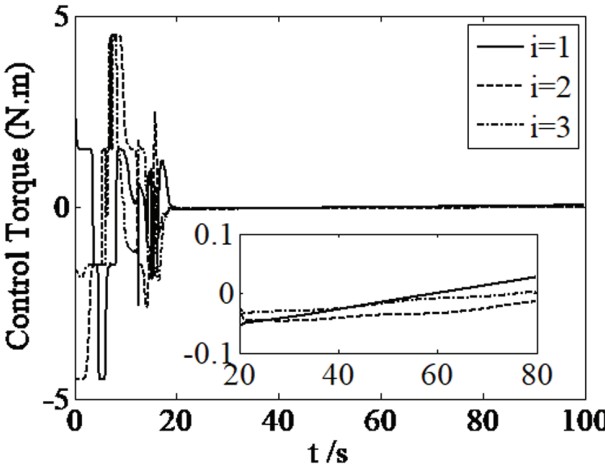

**Fig 14. Curves of control torque.**

## Author contributions

**Writing – original draft:** Yanmin Ren.

**Writing – review & editing:** Aijia Xing.

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
