## [Decision Letter · Decision Letter 0]

13 Jan 2025

PONE-D-24-51147Finite-time attitude tracking control for spacecraft based on backstepping method with input saturationPLOS ONE

Dear Dr. Ren,

Thank you for submitting your manuscript to PLOS ONE. After careful consideration, we feel that it has merit but does not fully meet PLOS ONE’s publication criteria as it currently stands. Therefore, we invite you to submit a revised version of the manuscript that addresses the points raised during the review process.

**ACADEMIC EDITOR: For this submission, I invited three reviewers. However, only one reviewer responded and submitted their review. To ensure a timely evaluation in consideration of the authors, I decided to proceed with this decision based on the single review received, without seeking additional reviewers. **

**The reviewer raised various comments regarding the presentation and the technical requirements of this paper. These comments are critical, and the authors should address them thoroughly. Based on my evaluation, this paper addresses the attitude tracking control for spacecraft with input saturation. However, the input saturation control problem has been widely considered in the control literature. Relevant works include Robust consensus control of second-order uncertain multiagent systems with velocity and input constraints, controllers for multiagent systems with input amplitude and rate constraints and their application to quadrotor rendezvous. The authors should provide a comparison with these studies to highlight distinctions.**

**In addition, there are some typos scattered throughout this paper, such as (23). Below (4),  "Where" should be corrected to  "where".**

We look forward to receiving your revised manuscript.

Kind regards,

Gang Wang

Academic Editor

PLOS ONE

Journal Requirements:

Bounded finite-time attitude tracking control for rigid spacecraft via output feedback - https://doi.org/10.1016/j.ast.2017.01.020

Finite-time output feedback attitude control for rigid spacecraft under control input saturation - 10.1016/j.jfranklin.2016.08.013

In your revision ensure you cite all your sources (including your own works), and quote or rephrase any duplicated text outside the methods section. Further consideration is dependent on these concerns being addressed.

3. Please remove your figures from within your manuscript file, leaving only the individual TIFF/EPS image files, uploaded separately. These will be automatically included in the reviewers’ PDF.

Additional Editor Comments :

AE Report: For this submission, I invited three reviewers. However, only one reviewer responded and submitted their review. To ensure a timely evaluation in consideration of the authors, I decided to proceed with this decision based on the single review received, without seeking additional reviewers. 

The reviewer raised various comments regarding the presentation and the technical requirements of this paper. These comments are critical, and the authors should address them thoroughly. Based on my evaluation, this paper addresses the attitude tracking control for spacecraft with input saturation. However, the input saturation control problem has been widely considered in the control literature. Relevant works include Robust consensus control of second-order uncertain multiagent systems with velocity and input constraints, controllers for multiagent systems with input amplitude and rate constraints and their application to quadrotor rendezvous. The authors should provide a comparison with these studies to highlight distinctions.

In addition, there are some typos scattered throughout this paper, such as (23). Below (4),  "Where" should be corrected to  "where".

Reviewers' comments:

Reviewer's Responses to Questions

**Comments to the Author**

1. Is the manuscript technically sound, and do the data support the conclusions?

Reviewer #1: Yes

2. Has the statistical analysis been performed appropriately and rigorously? 

Reviewer #1: Yes

3. Have the authors made all data underlying the findings in their manuscript fully available?

Reviewer #1: Yes

4. Is the manuscript presented in an intelligible fashion and written in standard English?

Reviewer #1: Yes

5. Review Comments to the Author

Reviewer #1: In this paper, the finite-time attitude tracking control problem for spacecraft based on the backstepping method is addressed.

Some comments are given to improve the paper quality.

A. There are some grammar errors in the text. For example, in page 2, the main contributions of the results in this paper are: (i) The first controller can inhibit external disturbances requiring no information about them.

B. In Nonlinear model and problem formulation, the problem description is not clear enough.

C. How to makes the attitude error e always satisfy ||e||< 1?

D. The article does not provide detailed technical means for handling input saturation. From the definition of Eq.31, how to ensure that control signals do not enter saturation illegally?

E. How to show the convergence of the system signals in the simulation? More comparison results should be given to show the advantage of this work.

F. Many works have been done to deal with the input saturation issues, such as event-triggered adaptive neural network tracking control for uncertain systems with unknown input saturation based on command filters, and adaptive command filtered backstepping tracking control for auvs considering model uncertainties and input saturation. So, how to demonstrate the innovation of this article.

6. PLOS authors have the option to publish the peer review history of their article (what does this mean?). If published, this will include your full peer review and any attached files.

Reviewer #1: No

---

## [Author Response · Author response to Decision Letter 1]

26 Mar 2025

All questions have been modified where highlighted in the text, see Response to Reviewers for more details.

---

## [Decision Letter · Decision Letter 1]

28 Apr 2025

PONE-D-24-51147R1Finite-time attitude tracking control for spacecraft based on backstepping method with input saturationPLOS ONE

Dear Dr. Ren,

Thank you for submitting your manuscript to PLOS ONE. After careful consideration, we feel that it has merit but does not fully meet PLOS ONE’s publication criteria as it currently stands. Therefore, we invite you to submit a revised version of the manuscript that addresses the points raised during the review process.

**ACADEMIC EDITOR: **

One review has been received for the revised paper. The reviewer remains unsatisfied with the revision and raises the following concerns: (1) the manuscript still contains numerous grammatical errors, (2) the benefits of the proposed approach compared to existing work are not clearly articulated, and (3) the paper lacks reliable comparative results.

In addition, I find that the authors have not addressed my previous concerns satisfactorily. In particular, the issue of input saturation control has been extensively studied in the control literature. Relevant works include "Robust consensus control of second-order uncertain multi-agent systems with velocity and input constraints," as well as studies on controllers for multi-agent systems with input amplitude and rate constraints and their application to quadrotor rendezvous. These references should be included in the revised manuscript to provide a more balanced and fair justification of the proposed approach.

We look forward to receiving your revised manuscript.

Kind regards,

Gang Wang

Academic Editor

PLOS ONE

Additional Editor Comments:

One review has been received for the revised paper. The reviewer remains unsatisfied with the revision and raises the following concerns: (1) the manuscript still contains numerous grammatical errors, (2) the benefits of the proposed approach compared to existing work are not clearly articulated, and (3) the paper lacks reliable comparative results.

In addition, I find that the authors have not addressed my previous concerns satisfactorily. In particular, the issue of input saturation control has been extensively studied in the control literature. Relevant works include "Robust consensus control of second-order uncertain multi-agent systems with velocity and input constraints," as well as studies on controllers for multi-agent systems with input amplitude and rate constraints and their application to quadrotor rendezvous. These references should be included in the revised manuscript to provide a more balanced and fair justification of the proposed approach.

Reviewers' comments:

Reviewer's Responses to Questions

**Comments to the Author**

1. If the authors have adequately addressed your comments raised in a previous round of review and you feel that this manuscript is now acceptable for publication, you may indicate that here to bypass the “Comments to the Author” section, enter your conflict of interest statement in the “Confidential to Editor” section, and submit your "Accept" recommendation.

Reviewer #1: (No Response)

2. Is the manuscript technically sound, and do the data support the conclusions?

Reviewer #1: (No Response)

3. Has the statistical analysis been performed appropriately and rigorously? 

Reviewer #1: (No Response)

4. Have the authors made all data underlying the findings in their manuscript fully available?

Reviewer #1: (No Response)

5. Is the manuscript presented in an intelligible fashion and written in standard English?

Reviewer #1: (No Response)

6. Review Comments to the Author

Reviewer #1: Although the paper answered some of the questions, the quality of the paper did not improve significantly. Specifically manifested as

1. The paper still contains a large number of grammar errors

2. Although the control strategy used in this paper can handle input saturation, it will result in the controller being particularly conservative, especially when considering the values of k1 and k2. What are the benefits of this approach compared to existing achievements

3. The paper still does not provide reliable comparative results, and more references need to be added to explain the research purpose

7. PLOS authors have the option to publish the peer review history of their article (what does this mean?). If published, this will include your full peer review and any attached files.

Reviewer #1: No

---

## [Author Response · Author response to Decision Letter 2]

16 May 2025

Reply (1):

According to the editor’s suggestion, the part of grammatical errors has been revised, and the specific types of corrected errors are related to tense unification (e.g., develop→developed) and correlative collocation (e.g., the \ a), as shown in the green-highlighted parts of the text.

Reply (2):

At the suggestion of the editors, a corresponding narrative has been included in the article in the yellow highlighted sections on page 1, paragraph 2, page 2, paragraph 1 and page 2, paragraph 2.

Reply (3):

At the suggestion of the editors, some references have been added to question (2) as a means of supporting the thesis statement of the paper, as detailed in the article in the yellow highlighted sections on page 1, paragraph 2, page 2, paragraph 1 and page 2, paragraph 2. And this paper combines the hypercurved tangent function with the auxiliary system to design a finite-time input saturation controller. Although the introduction of parameters makes the system response speed slightly slower than that under non-saturation conditions, showing a certain degree of conservatism, compared with the high-frequency jitter caused by the sign function (such as SGN) or piecewise saturation function used in some existing finite-time control methods, the controller in this paper maintains continuity in design and effectively avoids high-frequency switching and jitter of the control signal. More importantly, under the condition of limited physical output, the controller can still achieve high-precision attitude tracking within 20 seconds without any overshoot, showing good robustness and adaptability, ensuring good convergence performance when disturbances exist, output limitations and controller continuity coexist, and has significant advantages.

---

## [Decision Letter · Decision Letter 2]

27 May 2025

Finite-time attitude tracking control for spacecraft based on backstepping method with input saturation

PONE-D-24-51147R2

Dear Dr. Ren,

We’re pleased to inform you that your manuscript has been judged scientifically suitable for publication and will be formally accepted for publication once it meets all outstanding technical requirements.

Kind regards,

Gang Wang

Academic Editor

PLOS ONE

Additional Editor Comments (optional):

The reviewer is satisfied with this revision and recommends that the paper can be accepted. Therefore, I recommend acceptance of the paper.

Reviewers' comments:

Reviewer's Responses to Questions

**Comments to the Author**

1. If the authors have adequately addressed your comments raised in a previous round of review and you feel that this manuscript is now acceptable for publication, you may indicate that here to bypass the “Comments to the Author” section, enter your conflict of interest statement in the “Confidential to Editor” section, and submit your "Accept" recommendation.

Reviewer #1: All comments have been addressed

2. Is the manuscript technically sound, and do the data support the conclusions?

Reviewer #1: Yes

3. Has the statistical analysis been performed appropriately and rigorously? 

Reviewer #1: Yes

4. Have the authors made all data underlying the findings in their manuscript fully available?

Reviewer #1: Yes

5. Is the manuscript presented in an intelligible fashion and written in standard English?

Reviewer #1: Yes

6. Review Comments to the Author

Reviewer #1: (No Response)

7. PLOS authors have the option to publish the peer review history of their article (what does this mean?). If published, this will include your full peer review and any attached files.

Reviewer #1: No

---

## [Editor Report · Acceptance letter]

PONE-D-24-51147R2

PLOS ONE

Dear Dr. Ren,

I'm pleased to inform you that your manuscript has been deemed suitable for publication in PLOS ONE. Congratulations! Your manuscript is now being handed over to our production team.

Kind regards,

on behalf of

Dr. Gang Wang

Academic Editor

PLOS ONE